# Monitoring Plant Height and Spatial Distribution of Biometrics with a Low-Cost Proximal Platform

**DOI:** 10.3390/plants13081085

**Published:** 2024-04-12

**Authors:** Giovanni Bitella, Rocco Bochicchio, Donato Castronuovo, Stella Lovelli, Giuseppe Mercurio, Anna Rita Rivelli, Leonardo Rosati, Paola D’Antonio, Pierluigi Casiero, Gaetano Laghetti, Mariana Amato, Roberta Rossi

**Affiliations:** 1School of Agriculture, Forestry, Food and Environmental Sciences, University of Basilicata, 85100 Potenza, Italy; gianfranco.bitella@gmail.com (G.B.); bochicchiorocco@gmail.com (R.B.); donato.castronuovo@unibas.it (D.C.); stella.lovelli@unibas.it (S.L.); giuseppe.mercurio@unibas.it (G.M.); annarita.rivelli@unibas.it (A.R.R.); leonardo.rosati@unibas.it (L.R.); paola.dantonio@unibas.it (P.D.); 2Department of European and Mediterranean Cultures, Environment, and Cultural Heritage, University of Basilicata, 85100 Potenza, Italy; pierluigi.casiero@unibas.it; 3Istituto di Bioscienze e BioRisorse (CNR-IBBR), Consiglio Nazionale delle Ricerche, 80055 Bari, Italy; gaetano.laghetti@ibbr.cnr.it; 4Research Centre for Animal Production and Aquaculture (CREA-ZA), Council for Agricultural Research and Economics, 81100 Bella Muro, Italy; roberta.rossi@crea.gov.it

**Keywords:** plant spatial variation, canopy height, stress response monitoring

## Abstract

Measuring canopy height is important for phenotyping as it has been identified as the most relevant parameter for the fast determination of plant mass and carbon stock, as well as crop responses and their spatial variability. In this work, we develop a low-cost tool for measuring plant height proximally based on an ultrasound sensor for flexible use in static or on-the-go mode. The tool was lab-tested and field-tested on crop systems of different geometry and spacings: in a static setting on faba bean (*Vicia faba* L.) and in an on-the-go setting on chia (*Salvia hispanica* L.), alfalfa (*Medicago sativa* L.), and wheat (*Triticum durum* Desf.). Cross-correlation (CC) or a dynamic time-warping algorithm (DTW) was used to analyze and correct shifts between manual and sensor data in chia. Sensor data were able to reproduce with minor shifts in canopy profile and plant status indicators in the field when plant heights varied gradually in narrow-spaced chia (R^2^ = 0.98), faba bean (R^2^ = 0.96), and wheat (R^2^ = up to 0.99). Abrupt height changes resulted in systematic errors in height estimation, and short-scale variations were not well reproduced (e.g., R^2^ in widely spaced chia was 0.57 to 0.66 after shifting based on CC or DTW, respectively)). In alfalfa, ultrasound data were a better predictor than NDVI (Normalized Difference Vegetation Index) for Leaf Area Index and biomass (R^2^ from 0.81 to 0.84). Maps of ultrasound-determined height showed that clusters were useful for spatial management. The good performance of the tool both in a static setting and in the on-the-go setting provides flexibility for the determination of plant height and spatial variation of plant responses in different conditions from natural to managed systems.

## 1. Introduction

Plant height is the result of many genetic, environmental, and management factors and is one of the most important traits in plant ecology [1]. It has been discussed in terms of strategy linked to latitude and correlations with environmental variables. It affects the ability of species and individuals to compete for light and, therefore, to acquire carbon through photosynthesis and is a determinant of water evapotranspiration and seed dispersion [2,3]. A general relationship between height and biomass of vegetation across a range of ecosystems has been proposed [4] as a way to assess aboveground carbon stocks in natural systems. The height of plants is a very useful parameter in agronomy; it is a commonly recognized indicator of crop growing status. It is related to crop yield [5] and is the most relevant parameter for the fast determination of crop response to management and crop variability alone or in conjunction with other plant characteristics [6]. Phenotyping for the selection of high-yielding or stress-resistant genotypes uses plant height as one of the fast indicators of plant response. In wheat breeding programs, the importance of plant height has long been recognized [7]. Height variability at early stages of development could be used as a proxy of crop establishment and productive performances [8]. Monitoring plant growth at a high spatial density is essential in precision agriculture [9].

However, measuring plant height repeatedly during the season and at high resolution is a long and expensive task if performed manually. Technologies currently used to provide this parameter are as follows: stereo vision, laser image detection and ranging (LiDAR), and ultrasonic sensors. Stereo vision is a method of distance measurement based on the different perspectives on the same scene [10,11]. Its use is in expansion, related to the increasing diffusion of drones, but requires expensive computation [12]. The LiDAR yields a distance measure based on the time of flight of a laser beam emitted by the sensor and captured back by a photodetector [13]. It has been used to estimate features of herbaceous vegetation like crop density [14] and weed infestation [15]. Terrestrial LiDAR offers high resolution and cover, but its cost is still high. Like the LiDAR, an ultrasonic sensor is capable of detecting the distance from an obstacle indirectly by measuring the time of flight (TOF) of an ultrasonic pulse emitted by the sensor and echoed back by an obstacle. Compared to LiDAR, ultrasonic sensors are less expensive and do not involve possible harm to the naked eye. LiDAR, on the other hand, can make measurements at a larger distance and features narrow point detection; this can be a pro as it gives a higher accuracy but also a con as given objects may be missed in a prospection. Ultrasonic sensors have been used in agriculture to provide cheap and repeatable distance information related to crop biomass [16], canopy density, to optimize pesticide applications [17], and for weed detection [18].

Ultrasonic devices are still underutilized in agriculture and can offer, at low cost, very useful information to researchers and farmers. Integrating crop sensors through open-source hardware projects might help fill this gap. Open hardware platforms are easy to realize and use; their low cost, coupled with the increasing amount of information sharing, supports the rapid development of new-generation, inexpensive, and very flexible scientific instrumentations [19,20]. Ultrasound-based high-throughput sensors or platforms have been proposed for use in different crops [21,22,23,24,25,26,27], mostly in the on-the-go mode.

Montazeaud and co-authors [28] recently proposed a low-throughput sensor intended for manual use for small-scale applications such as plot experiments where measurements of the height of individual plants are needed. They tested it on single-plant measurements on sorghum. The device is easy to use although limited to static measurements.

We aimed to devise a low-cost system for measuring plant height which could be used in a flexible way, both on-the-go and in static mode, and to prove its usefulness in different applications from single plants to plots and whole fields. We therefore aimed to test the device across a range of vegetation types, from crop row transects to whole-field mapping.

## 2. Results

### 2.1. Lab Tests

Figure 1 reports the results of laboratory testing. Ultrasound-derived height (hus) followed the contour of manually measured height (h) with a significant correlation (r = 0.86, *p*-value < 0.05). However, a slight shift in data was observed and the sensor overestimated the height of shortest targets by 0.89 cm on average (Figure 1).

The sample cross-correlation shows that the highest correlation (r = 0.88) occurs at lag 1. Hence, a one-lag shift between the two series slightly improved the overall correlation. The Dynamic Time Warping Normalized Distance (DTW) between sensor data and manually measured height was 0.39.

### 2.2. Faba Bean (Vicia faba L.)

Figure 2 reports ultrasound-measured heights as a function of ground truth data for *Vicia faba* L. Plant heights were significantly (*p* < 0.05) related to sensor-measured height data with high R^2^ values (0.96 for the regression of pooled data), which were not significantly improved by dividing var. *major* from var. *minor* data. Values for single varieties were substantially aligned (Figure 3b,c).

### 2.3. Chia (Salvia hispanica L.)

In *Salvia hispanica* field transect on rows with 5-cm plant spacing (Figure 3a,b) the crop height pattern obtained with manual measurements (Figure 3a blue dots) was closely reproduced by ultrasound-derived height (Figure 3a orange dots) in the zone between 40 and 260 cm of distance along the transect, but underestimated plant height of about 4.4 cm. At the transect edges (About 40 cm from each end of the row) sensor data failed to reproduce the height profile and differences were higher. The sample cross-correlation was maximum at lag 1 therefore a second series of height data derived from shifting ultrasound data by one lag was created (Figure 3a black dots). The use of a dynamic time warping algorithm yielded a DTW distance of 3.38, and data corrected for DTW are shown in Figure 3a as black dots.

Across the whole transect, including edges, regression between hus and ground truth (h) values (Figure 3b) yielded a significant relationship (*p* < 0.05) both at 0 (raw data) and 1 lag (respectively, orange and black dots and regression lines). The non-significant intercept was removed, and the regression line was forced to pass through the origin yielding high values of R^2^ even for raw data (almost 0.97 in Figure 3b, orange dots), and slightly improved (0.98) for data shifted by 1 lag (Figure 3b black dots) and DTW-warped (Figure 3b green dots).

In the case of plant spacing of 30 cm (Figure 3c,d), hus data (orange dots) follow the overall crop profile but fail to reconstruct short-scale variability in plant height, and especially bare-soil points (e.g., at 50, 85 and 115 cm along the transect) and a whole bare-soil stretch between cm 165 and 215 along the transect. The maximum correlation is found at lag 2 but even data shifted in 2 lags (black dots in Figure 3c,d) do not reproduce short-scale variability and relevant very low height or zero values. The use of a dynamic time warping algorithm yielded a DTW distance of 7.94, and a better reproduction of the pattern of manual data is found with a data series calculated with dynamic time-warping (Figure 3c green dots). However, very low and zero values are not reproduced. The correlation between manual and sensor data is lower for plants spaced at 30 cm than for those at 5 cm but is still significant (*p*-value = 0.04). The univariate regression model still explains a considerable amount of the total variability if the line is forced to pass through the origin (R^2^ = 0.57 for data after 2-lag shifting and 0.66 for data after DTW correction). Sensor data overestimate height by about 14.0 cm on average in this transect after 2-lag shifting (black dots) and 7.3 cm after correction with DTW (green dots).

### 2.4. Alfalfa (Medicago sativa L.)

Figure 4 shows maps of ultrasound-measured canopy height at three plant heights (Figure 4) in an alfalfa (*Medicago sativa* L.) field.

Values of ultrasound-measured plant height ranged between 2.51 and 12.07 cm on the first date, between 3.97 and 16.75 am on the second, and between 8.02 and 57.76 cm on the third. Plant height showed an unimodal frequency distribution at all three dates (Figure 4 right), but values were not spatially distributed at random in the field: maps show that clusters of low values (blue areas) were found in the top and left field zones, and high values (red areas) in the center of the field in the first two dates, whereas at the third date high values (red areas) were found at the top edge and in a strip inside the field, and low values (blue areas) in the center, and frequencies were slightly skewed to the left: height values lower than the mode were less frequent.

Figure 5, Figure 6 and Figure 7 report data collected in the ground-truth areas and summary statistics for plant biometrics and sensor indices are reported in Table 1.

Figure 5 shows the relationships between manually measured plant height (h) and other biometrics. Significant (*p* < 0.05) linear regressions with R^2^ values higher than 0.9 were found with fresh and dry mass and LAI (Figure 5a–c), and a power regression with leaf/stem ratio (Figure 5d).

The same regression models were fitted for relationships between normalized difference vegetation index (NDVI) and other biometrics (Figure 6), which were all significant (*p* < 0.05) and linear for fresh and dry mass and LAI (Figure 6a–c), and power regression for leaf/stem ratio (Figure 6d). Values of R^2^ were lower than those found for regressions with plant height (Figure 5).

Figure 7 shows the relationships between manually (h) and ultrasound-measured (hus) plant height (Figure 7a) and between hus and other biometrics (Figure 7b–f). For all tested models, the highest R^2^ values were found for linear regressions, and they were all significant (*p* < 0.05). Sensor-derived height (hus) was significantly related to ground-truth plant height (h) (Figure 7a), and the model explained 89 to 98% of the variability if the intercept is kept or removed, respectively. The hus variable was also significantly related to NDVI with a 0.72 R^2^ value (Figure 7b). Regressions of LAI and biomass with hus (Figure 7c–e) were characterized by R^2^ values higher than those of the same variables with NDVI (Figure 6) and lower than those with h (Figure 5). The regression of leaf/stem ratio with hus (Figure 7f) was linear and had a higher R^2^ than those with h (Figure 5d) and NDVI (Figure 6d).

### 2.5. Wheat (Triticum durum Desf.)

Results from the wheat experimental field are reported in Figure 8 and Figure 9. Maps of sensor values across the experimental field at the end of tillering are reported in Figure 8. A number of 481 to 493 values were acquired in each plot and the maps of hus (Figure 8a) and NDVI (Figure 8d) show differences between plots of the C and W treatments. In the C plot, the blue color dominates, corresponding to low values (color-scales for hus and NDVI at Figure 8b and e respectively), whereas in the W plots, the dominant color is red, corresponding to high values. Maps also show within-plot variability with different shades of blue in the C plots and colors from green to black in the W plots. Frequency distributions of values for the whole field (Figure 8b,e) were bimodal, corresponding to the different modes of the treatments, respectively for hus 4.14 cm in C and 12.98 cm in W, and for NDVI values of 0.11 in C and 0.27 in W. Average values of the W plots were more than twice those of the C plots for ìboth hus and NDVI. Data from single transects at the same phenological stage, when maps were made (end of tillering), are summarized across the whole field in Table 2.

Figure 9 reports averages of plant heights from cut (C) and uncut (W) treatments. Values were significantly (*p*< 0.05) different throughout the growth cycle for both h and hus.

At the end of tillering (Figure 9a) sensor data (hus) were not significantly different from manually measured heights for both cut and uncut wheat, whereas hus was significantly lower than h at the booting stage (Figure 9b) and significantly higher within cutting treatment at grain filling (Figure 9c). Nevertheless, the overall linear regression between h and hus on data from the three phenologoical stages was significant (*p* < 0.05) and explained 96% of the variability (hus = 1.0031 h + 1.68 R^2^ = 0.96).

## 3. Discussion

In our data the bivariate relationship between ultrasound-measured height and manual data was always significant, therefore our sensor can be considered a useful tool, and hus can be taken as a proxy of plant height across a range of canopy types, from rows of spaced plants (e.g., *S. hispanica*) to canopies spread over whole fields (e.g., *M. sativa*), and this may be extended to different conditions of crops or natural vegetation.

This agrees with data in the literature where ultrasound measurements provide reliable plant height data in a range of crops, and in static or on-the-go modes [21,22,23,24,25,26,27,28].

Our sensor performed with different accuracy in the different modes we tested: the regression models explained up to 99% of the variability in wheat at the end of tillering, but as little as 54 to 66% of the variability when used on-the-go on wide-spaced chia plants (Figure 3c,d). Sharp variation of the target’s contour are not completely caught as in lab setting (Figure 1) or at the edge of chia rows (Figure 3) or in case of wide-spaced chia plants (Figure 3c,d) where the sensor was much less accurate in reconstructing short scale variability in plant height and did not pick up narrow canopy voids. This is due to interference of neighboring features within the sensor’s field of view and has been documented in the literature (e.g., [15]). More gradual height changes were reproduced by the sensor with higher accuracy, as in the chia transect at 5-cm plant spacing where changes in height corresponded to gradual variations in chia plant tops, or to the presence of shorter *Amaranthus retroflexus* L. plants which represented the main segetal species in the field.

No systematic error was recorded in some cases for both static or on-the go measurements (e.g., in faba bean, Figure 2 or alfalfa, Figure 7a or wheat during tillering, Figure 9a). Nevertheless they emerged in other cases: under-or over-estimation and minor shifts were found in laboratory measurements (Figure 1) and in chia (Figure 3a,d). A measure of misalignment was obtained by cross correlation or using a dynamic time warping algorithm. After quantifying misalignments, we used cross-correlation or dynamic time-warping distance (DTW) to shift hus measurements one of one or few lags and obtained a better correspondence of sensor and measured data (e.g., Figure 3c,d for chia plants spaced at 30 cm). Cross correlation is a feature-based measure of similarity between data series [29] and was used in this work to shift sensor data with respect to manual data in order to improve data matching. The need to shift data series can be ascribed to a lag time in data collection or ground-positioning system (gps) data recording linked to the speed of the moving sensor. The dynamic time warping algorithm is a shape-based procedure to measure similarity between data sequences, through quantifying the distance between similar elements in different series of data [29]. It originates from time-series analysis [30] but can be applied to shape-matching, and in particular to find out if similarity or matching in shape can be found between two data sequences which are out of phase. The Dynamic Time Warping (DTW) distance is therefore a measure of the level of dissimilarity between data series. In our data the DTW-normalized distance was 0.39 in lab setting, 3.38 in the chia row at 5-cm spacing and 7.94 in the chia row with 30-cm spacing, this showing an increasing level of dissimilarity between sensor and manually determined height.

Even where alignment between sensor and manual data series is improved through shifts based on cross-correlation or analyzed through time-warping, the problem of over- or under- estimation persists. In our data for instance the chia transect with plants spaced 30 cm (Figure 3c,d) showed overestimation of 7.3 to 14.4 on average, respectively for data after shifting based on DTW or cross-correlation. This was due to inability to detect very low or zero values when interspersed with tall plants. In *Salvia hispanica* (Figure 3) hus values closely reproduced plant height except for at the edges of measured transect. In this experiment the edges of transects corresponded to plot edges, therefore to regions where bare soil and/or segetal species were found, with height different from chia. We hypothesize that in this case specific edge effects coupled with shifts and systematic errors in sensor measurements we recorded in this dataset may be due to an imperfect perpendicularity of the sensor, which may have picked up reflections from bare soil at the beginning of the transect, thus underestimating plant height, and reflections from the crop at the transect end, thus overestimating plant height.

Ultrasound measurements underestimated wheat height at booting (Figure 9b), and overestimated it at grain filling (Figure 9c) and this may be ascribed to general factors and specific issues linked to the method of height measurement. For the booting stage the flag leaf possibly was not a consistent enough target for the ultrasound beam to pick it up completely. This is consistent to the type of underestimation explained by Sui and Thomasson [31] as occurring when the sensor is not centered on the top or top-leaf of the plant (And this may happen frequently for on-the go measurements): in such cases the closest leaf that echoes back the signal would not be the highest. At the wheat booting stage (Figure 3c) a lower h is explained by the fact that it did not include awns as described in the Materials and Methods section.

In general, biases between manual and sensor measures could have been caused by several reasons ranging from sensor misalignment, to soil micro-topography and to the influence of temperature, which we didn’t consider in our analysis. Several factors can reduce ultrasonic accuracy: the presence of systematic features like ridges and furrows, the influence of air temperature and the inherent sensor transducer accuracy. Canopies are porous, hierarchical structures, therefore the echoing of the signal depends on leaf morphology, angle and canopy architecture. Also inconsistencies might be due to the multiple reflected signals within the signal field of view [31]. All these issues lead to a cumulative error, and must be taken into account if quantitative predictions are needed. Our data confirm that a crop- and even a crop-stage specific calibration would always be required [32,33]. Having experimented our sensor in a range of modes from static (faba bean) to on-the-go at the row (chia, wheat), plot (wheat, alfalfa) and field (alfalfa) scale and having found different merits and drawbacks of each model, we can add that sensors should be tested in different conditions and modes.

Nevertheless in our data even overall regression models encompassing different growth stages or crop varieties were significant and explained a large part of the variability (e.g., 96% in wheat across growth stages and in faba bean across botanical and varieties, accessions and commercial varieties) Also, even where the sensor systematically underestimated or overestimated canopy height, areas where canopy height changed could be well mapped (Figure 2, Figure 3a,b and Figure 4, Figure 5, Figure 6, Figure 7, Figure 8 and Figure 9), except for very narrow variations as in chia spaced plants (Figure 3c,d), and differences between experimental treatments were well quantified (Figure 8 and Figure 9).

In our measurement settings we aimed at different conditions: we tested it on broadleaf (e.g., faba bean) and narrowleaf (wheat), plants with large (e.g., chia) or small (e.g., alfalfa) laves and on a wide range of plant heights over which the sensor was tested, not only between but also within experiments, as quantified by high values of coefficients of variation. We had satisfactory to excellent agreement with ground-truth data, and this confirms the sensor is a good tool for different herbaceous vegetation types and can pick up field variability and possibly help in spatial applications. Applications in ecology include the spatial distribution of primary production and of plant light interception. A general relationship between height and biomass of vegetation across a range of ecosystems has been proposed [4] as a way to assess aboveground carbon stocks in natural systems, and our sensor may provide a flexible tool for height measurements over large areas or along transects.

Monitoring plant growth at a high spatial density is essential in precision agriculture [8] therefore height maps as those we show for alfalfa (Figure 4) or wheat (Figure 8) can be of assistance in precision farming operations.

Important crop biometrics other than manually measured height are well related to ultrasound measurements in our data, such as the leaf area index (R^2^ = 0.79 in wheat and 0.81 in alfalfa) and the NDVI (R^2^ = 0.98 in wheat and 0.72 in alfalfa). Plant height proved to be a better predictor of biomass than NDVI in alfalfa, where both h (Figure 5) and hus (Figure 7) showed higher R^2^ values in the bivariate regression models with fresh (Figure 5a and Figure 7d) and dry (Figure 5b and Figure 7e) mass than than NDVI (Figure 6a,b). As vegetative biomass is the commercial product of alfalfa, a map of biomass corresponds to a map of forage yield in this crop. Yield maps are crucial tools for driving precision farming operations [34], and given the high predictive value of hus over biomass (R^2^ = 0.81 for fresh and 0.84 for dry mass, Figure 7a,e), maps of ultrasound-determined height can be considered proxies of yield maps and used for driving delineation of uniform management zones in spatially-aware farming [35].

A relationship between ultrasound-measured height and biomass in forage crops has also been found by Fricke and co-authors [15] but with lower R^2^ than in our case. Other measurements on forage crops [36,37] were less accurate than in our case with R^2^ values between 0.7 and 0.8, but over a wide range of conditions, and results nare also discussed in terms of vehicle speed. Authors, though, stress higher accuracy compared to other non-desctructive methods for predictiong forage biomass like the rising-plate meter [37] and airborne or satellite-based radiometric methods constrained by equipment cost, expertise and/or meteorological conditions.

In our data hus was significantly (*p* < 0.05) and strongly related to LAI (R^2^ = 0.79 for wheat to 0.81 for alfalfa), therefore ultrasound-measured height may also be used as a fast measurement for agronomic decisions related to leaf area as Leaf Area Index is a crucial parameter for crop modeling and input management, especially irrigation (e.g., [38]. In the case of alfalfa we also found an inverse relationship of height with the leaf to shoot ratio, which is a parameter of forage quality, and this widens the range of agronomic decisions our sensor may assist with.

The board used for our sensor is especially amenable for flexible uses and therefore to address error sources linked to different settings (e.g., [39]) and for use in mono-sensors or for multi-sensor platform (e.g., [40]) Internet of Things applications given the different connectivity modes from Wi-Fi to Bluetooth Classic and Bluetooth Low Energy (BLE), Sim card or Ethernet support and additional interfaces, including UART, SPI, I2C, and ADC, enabling connection to a wide range of peripheral devices. Flexibility is also given by a high number of General-Purpose Input/Output pins and a large flash memory for programming and storage, and compatibility with other boards and development platforms.

## 4. Materials and Methods

### 4.1. Ultrasound Sensor Platform

The platform used in this study was composed of:(1)An ESP32 board (Espressif Systems, Singapore) with a dual-core microcontroller. Tensilica Xtensa 32-bit LX6 microprocessor with wireless connectivity Wi-Fi: 802.11 b/g/n/e/i (802.11n @ 2.4 GHz up to 150 Mbit/s) and Bluetooth: v4.2 BR/EDR and Bluetooth Low Energy (BLE). The current cost of an ESP32 board ranges from 1.75 to 8 Euros depending on the source.(2)An ultrasound sensor was an HC-SR04 (Picaxe, Revolution Education Ltd, Bathh, UK) transmitting at 40 KHz frequency, and operating between 3 and 400 cm of distance with accuracy of 3 mm with a cone of 45 degrees from the sensor. The HC-SR04 rapidly generates a series of ultrasound pulses which propagate in a straight line in front of the sensor. The ultrasounds hit an object in front of the sensor and are reflected back towards the sensor, which detects the time taken for the ultrasound pulses to travel from their source to the object and back. The sensor uses the elapsed time to calculate the distance between itself and the object as:
Distance = (Elapsed time × Speed of sound)/2

The current cost of an HC-SR04 ranges from 2 to 10 Euros depending on the source,

(3)A Zs-040 module which sends data via Bluetooth. This was added in order to simplify hardware and make data easily available in real time thanks to transmission to a PC or smartphone. The current cost ranges from 0.3 to 10 m Euros depending on the source.

Connections of electronic circuits are depicted in Figure 10.

All the electronic components were enclosed in a protective plastic case (current cost 5 Euros). The whole device can be easily powered through the USB-C port. We connected a 7860 mAh power bank, commonly used for charging smartphones, to the USB port (current cost 15 Euros), which allowed its use for several hours.

### 4.2. Data Collection

Ultrasound data were collected in both lab (static measures) and field setting (static or on-the-go).

Static measurements were made after mounting the sensor on a pole at a fixed distance from the ground (e.g., Figure 11a).

On-the-go measurements were collected connecting a differential GPS and the sensor was mounted either on a quad (Figure 11b) or on straddle wheeled chassis (Figure 11c), kept at a fixed distance above the ground (sensor distance = hs) pointing vertically down to the row, and towed manually across the field.

### 4.3. Sensor Testing

To test the ultrasonic sensor accuracy data have been ground-calibrated taking manual measurements of target height.

In all tests the height of target objects or vegetation from ultrasound measurements (hus) was calculated as:hus = hs − usd (cm)
where
hs = height of sensor from the ground
usd = ultrasound-measured distance between target objects or vegetation and sensor.

The sensor was tested in conditions of different complexity:

#### 4.3.1. Lab Test

The sensor was mounted on a wheeled chassis at the distance of 30 cm from the floor where dark boxes of heights from 2.5 to 8 cm were aligned in a transect, and the chassis was shifted along the transect at an average speed of 0.1 m s^−1^. The ultrasonic sensor triggered 5 measures per second, e.g., a measurement every 0.02 m.

Field tests were conducted at four sites in Southern Italy as indicated in Figure 11 on crops of different geometry and namely:

#### 4.3.2. Faba Bean (*Vicia faba* L.)

The field was located at Lavello (Italy (Lat. N 41°07′28.49″, Long E 15°91′89.59″) and texture data were: 31% sand, 33% silt, 36% clay, and organic matter amounted to 16.4 g kg^−1^. Measurements were made in static mode on row-planted single plants of faba bean (*Vicia faba* L.). For the botanical variety *Vicia faba* (L.) var. *maior* (Harz) Beck, we used the commercial variety Aguadulce and accessions sourced from the germplasm collection of the Institute of Biosciences and Bioresources (IBBR) of the Italian National Research Council (CNR) in Bari, Italy:Vma1 = Accession number 112906 from USA Vma1Vma2 = Accession number 103235 from ItalyVma3 = Accession number 107620 from Greece.Vma4 = Accession number 106374 from Algeria

For the botanical variety *Vicia faba* L. var. *minor* (Harz) Beck we used the commercial variety Prothabat and accessions sourced from the germplasm collection of the Institute of Biosciences and Bioresources (IBBR) of the Italian National Research Council (CNR) in Bari, Italy:Vmi1 = Accession number 113620 from GermanyVmi2 = Accession number 113620 from GermanyVmi3 = Accession number 109322 from EthiopiaVmi4 = Accession number 118952 from Afghanistan

Measurements were made for *Vicia faba* (L.) var. *maior* at phenological stage 20 (no side shoots) for the BBCH scale and for *Vicia faba* (L.) var. *minor* at phenological stage 21 (Beginning of side shoot development: first side shoot detectable) for the BBCH scale. Acquisitions were made in static mode with the ultrasound sensor mounted on a pole and placed above single plants at 80 cm distance from the ground (Figure 10). For ground-truth height measurements (h) were made manually with a rigid measuring tape on the same plant where ultrasound data were acquired. The variable h = plant height was the distance between the ground and the top of the uppermost plant structure.

#### 4.3.3. Chia (*Salvia hispanica* L.)

The field was located at Masserie Saraceno (Atella, Italy, Lat. N 40°51′37.59″, Long. E 15°38′49.43″) on loam soil with the following characteristics: sand 43.6%, silt 34.2%, clay 22.1%. The broad leaf crop *Salvia hispanica* L. was sown in rows with plant spacing of 5 cm or 30 cm on the row. One transect for each of the plant spacing treatments was chosen based on the presence of gradients of plant height and/or areas of bare soil along the row. Sensor measurements were acquired on-the-go with the ultrasound sensor mounted on straddle wheeled chassis and moved at an average speed of 0.25 m s^−1^. The ultrasonic sensor triggered 5 measures per second, e.g., a measurement every 0.05 m. Ground-truth measurements were made manually with a rigid measuring tape every 0.05 m along the transect. This corresponded to plant tops or lateral leaves or bare soil, or to *Amaranthus retroflexus* L. plants which represented the main segetal species in the field and were found along chia rows or beyond the edges of the plot. Data points were geo-referenced with a RTK GPS k9t (Kolida Instrument Co., Ltd, Guangzhou, China) with an accuracy of ±2 cm. Sensor and manual measurements were paired after retrieving from the sensor dataset observations at the closest location to measured data.

#### 4.3.4. Alfalfa (*Medicago sativa* L.)

Measurements were made in a 7-ha alfalfa (*Medicago sativa* L. cv. Altiva) stand in Palomonte (Italy, Lat N 40°61′39.52″ Long E 15°30′32.64″) at 210 m asl. The soil was classified as a Typic Eutrudept fine, mixed, thermic Calcaric Cambisols (Soil Survey Staff, 1999; IUSS Working Group WRB, 2006). The average soil texture within the first 0.5 m layer was 41.29% sand, 17.14% silt, 41.57% clay; the average soil organic matter content was 26 g kg^−1^. The stand was planted at a seeding rate of 40 kg ha^−1^.

Crop height was mapped on-the-go after mounting the ultrasound sensor on a quad at 0.6 m distance from the ground. Data points were geo-referenced with a RTK GPS k9t (Kolida Instrument Co., Ltd., Guangzhou, China) with an accuracy of ±2 cm. Maps were obtained at three times corresponding to three different heights of the alfalfa stand: a. maximum plant height 12.07 cm; b. maximum plant height 16.75 cm; c. maximum plant height 57.56 cm. On the latter date ground truth data for sensor testing were taken in 16 areas in the field which were chosen across the whole range of crop height values obtained from on-the-go maps with a surface-response-sampling method [41]. The following plant biometric and radiometric measurements were made:

##### NDVI

A radiometric Greenseeker^®^ (Trimble, Sunnyvale, CA, USA) sensor was used to meaasure reflected radiation in the red (~660 nm) and near infrared (~770 nm) wavelengths for the calculation of the Normalized Differences Vegatation Index (NDVI):NDVI = (NIR − VIS)/(NIR) + (VIS)
where
NIR = reflectance in the infrared band (~770 nm)
VIS = reflectance in the red band (~660 nm).

##### Leaf Area Index

Leaf area index (LAI m^2^ m^−2^) was measured with a LI-COR 2200c (LI-COR, Lincoln, NE, USA) field leaf-area meter.

##### Vegetation Height

Vegetation height was measured with a rigid measuring tape as the distance from the ground of a 10-cm diameter disk mounted to a stick and placed on top of the canopy every 0.1 m on a 0.5 × 0.5 m area.

##### Biomass

Above-ground plant parts were harvested on 0.5 × 0.5 m areas and weighed fresh and after oven-drying at 65 °C until constant weight.

#### 4.3.5. Wheat (*Triticum durum* Desf.)

Measurements were made in a hard wheat (*Triticum durum* Desf. cv. Tirex) stand in Genzano di Lucania (Italy Lat N 40°49′24.9″ Long E 16°05’33.8″. The average soil texture within the first 0.5 m layer was: sand 18.8%, silt 52.7%, clay 28.5%; organic matter 20.17 g kg^−1^. The stand was planted in rows at a distance of 15 cm and at a seeding rate of 230 kg ha^−1^.

Six plots of 2 × 3 m were set up to compare the height of wheat plants grown for grain (W = whole) with that of plants grown as a dual-purpose crop and, therefore, cut at the end of tillering at 0.07 m from the ground level (C = cut). Measurements were made at the end of tillering-beginning of stem elongation, booting, and grain filling stages (respectively stages 30, 41, 71) of the Zadoks scale [42]. Sensor measurements were acquired on the go along a row of each of the plots with the ultrasound sensor mounted on a straddle wheeled chassis. At the end of tillering and booting, plant height (h) was measured with a rigid measuring tape, which was the distance from the ground of a 10-cm diameter disk mounted to a stick and placed on top of the canopy. At grain filling, plant height (h) was measured manually with a rigid measuring tape as the distance from the ground to the top of the ear.

At the Zadoks stage, 30 ultrasound measurements were made along a row where LAI and NDVI were also measured with methods described in Section 4.3.3, and on the whole plots, where NDVI was also measured with methods described in Section 4.3.3.

### 4.4. Statistical Analysis

Data from the ultrasound sensor were analyzed in comparison with ground-truth height data and other plant biometrics with univariate regression models. On data from the wheat plot experiment, analysis of variance was conducted, and means were separated with the *post-hoc* test of Tukey. Where horizontal shifts of sensor data compared to manual data were found, we calculated feature-based and shape-based distance measures: respectively cross-correlation distance and Dynamic Time Warping normalized Distance (DTW) to account for possible misalignment between series [30]; DTW is based on the Euclidean distance computed after using dynamic programming to find the minimal path in a distance matrix between similar elements in two compared series [2]. All statistical analyses were performed within the R environment for statistical computing (version 3.1.2 [43]).

## 5. Conclusions

We devised and tested a low-cost platform for measuring plant height with an ultrasound sensor designed to be used in different modes, from static to on-the-go.

We generated point measurements with the static setting on faba bean, with a high regression coefficient where the ultrasound sensor proved able to reproduce height regardless of genotype and phenological stage, thus showing the potential of the platform for static single-plant measurements.

High regression coefficients were also shown in the on-the-go mode for settings with a quite continuous plant cover, such as narrow-spaced chia, wheat, and alfalfa, whereas sensor data were not able to closely reproduce crop height where sharp variations were found, such as in widely spaced chia.

We were able to quantify and correct some systematic errors, such as data misalignment with feature-based (cross-correlation) or shape-based (dynamic time-warping) measures of similarity between data series.

Ultrasound-measured plant height proved to be a better predictor than NDVI for plant biometrics relevant to water relations and yield behavior, such as Leaf Area Index and biomass in alfalfa.

Overall, ultrasound-measured height with our platform proved to be a fast and low-cost method of estimating crop parameters that is useful in ecological research or agriculture applications.

Important characteristics of our platform are that it is simple and flexible, given the possibility to be employed by users with different skills and inclinations for technology and in different settings, from mounted on simple poles for single-point static measurements to towed manually for the complete characterization of plots or rows in experiments. Further, the platform may be carried on vehicles for mapping large surfaces like open fields, prairies, and natural herbaceous vegetation sites, providing maps useful for spatial management and characterization of spatial variation of plant responses in different conditions from natural to managed systems.

One of the features of the platform’s ESP32 board is wide connectivity. Therefore, future developments may include the design of a multi-sensor platform with the same flexibility of use in different settings.

Future work should also focus on the analysis and correction of errors linked to field settings and modes of platform use, such as data misalignment.

## Figures and Tables

**Figure 1 plants-13-01085-f001:**
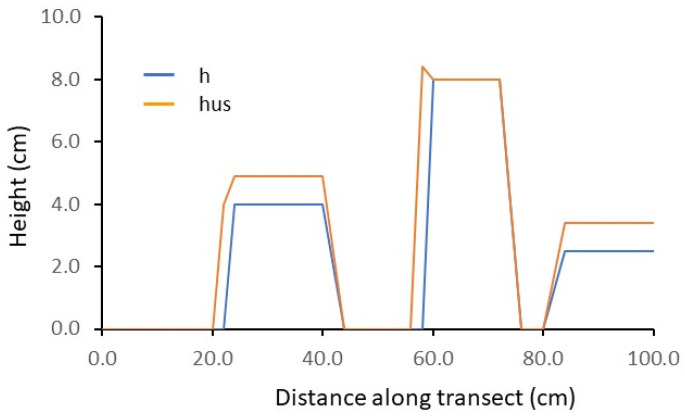
Comparison of sensor-determined and measured height of parallelepipeds in the laboratory. ultrasonic data (hus = orange line) overlaid by ground truth data (h = blue line).

**Figure 2 plants-13-01085-f002:**
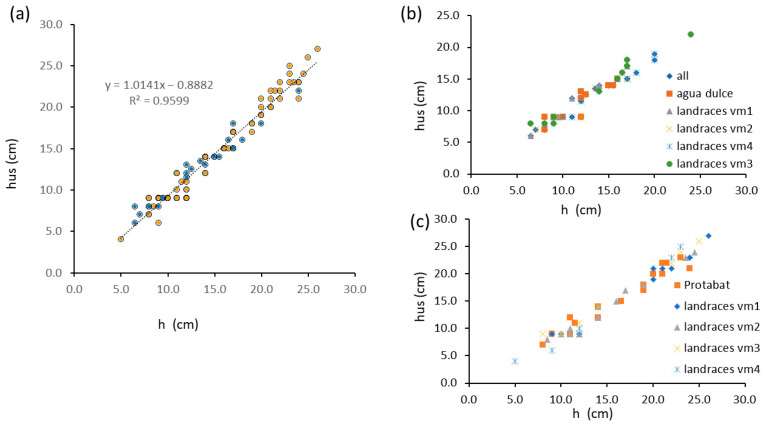
Height measurements in a *Vicia faba* L. field setting. (**a**) bivariate plot of ultrasound-measured heights as a function of ground truth data for *Vicia faba* L. var. *major* (blue dots) and *Vicia faba* L. var. minor (orange dots); (**b**) bivariate plot of ultrasound-measured heights as a function of ground truth data for *Vicia faba* L. var. *major* by variety; (**c**) bivariate plot of ultrasound-measured heights as a function of ground truth data for *Vicia faba* L. var. *minor* by variety.

**Figure 3 plants-13-01085-f003:**
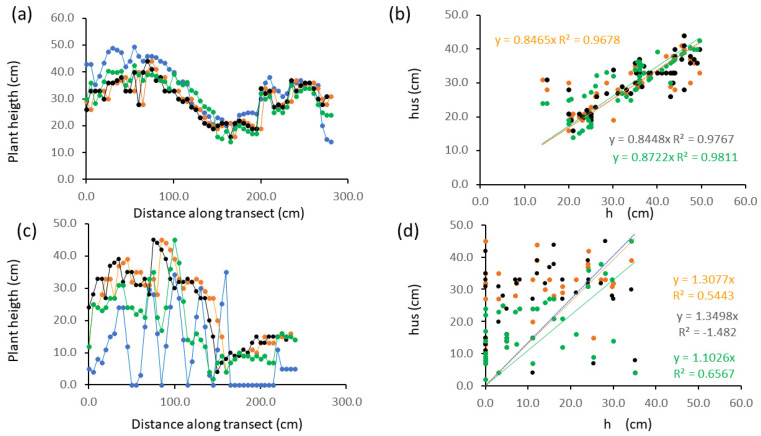
Height measurements in *Salvia hispanica* L. field setting. (**a**) row with plants spaced 5 cm: bivariate plot of ground truth data (blue dots), ultrasound-measured heights (orange dots) and ultrasound-measured heights shifted of 1 lag distance (black dots), data from dynamic time-warping (green dots); (**b**) row with plants spaced 5 cm: regression between measured height and ultrasound-measured height (orange dots and regression equations), ultrasound-measured height shifted of 1 lag distance (black dots and regression equations), and data from dynamic time-warping (green dots and regression equation); (**c**) row with plants spaced 30 cm: bivariate plot of ground truth data (blue dots) ultrasound-measured heights (orange dots), ultrasound-measured height shifted of 2 lag distance (black dots), data from dynamic time-warping (green dots); (**d**) row with plants spaced 30 cm: regression between measured height and ultrasound-measured height (orange dots) and ultrasound-measured height shifted of 2 lag distance (black dots and regression equations); data from dynamic time-warping (green dots and regression equations).

**Figure 4 plants-13-01085-f004:**
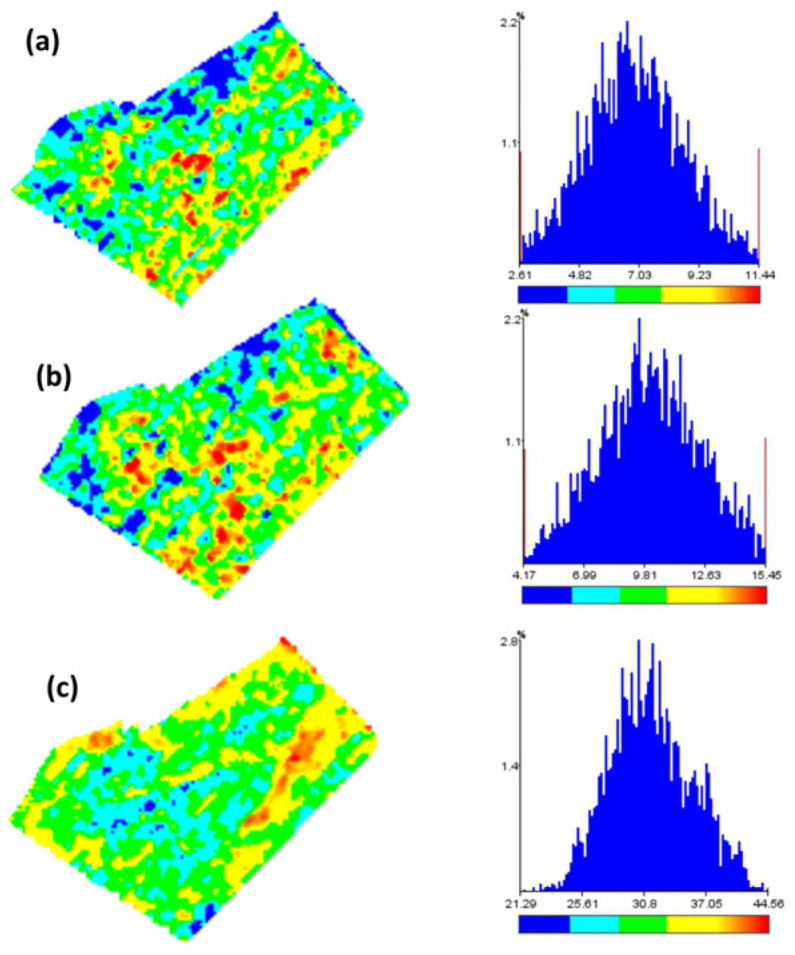
Maps of ultrasound-measured canopy height in a *Medicago sativa* L. field at (**a**) plant maximum height 12.07 cm; (**b**) maximum plant height 16.75 cm; (**c**) maximum plant height 57.56 cm. Left: plant height (hus) maps. Right: frequency distributions.

**Figure 5 plants-13-01085-f005:**
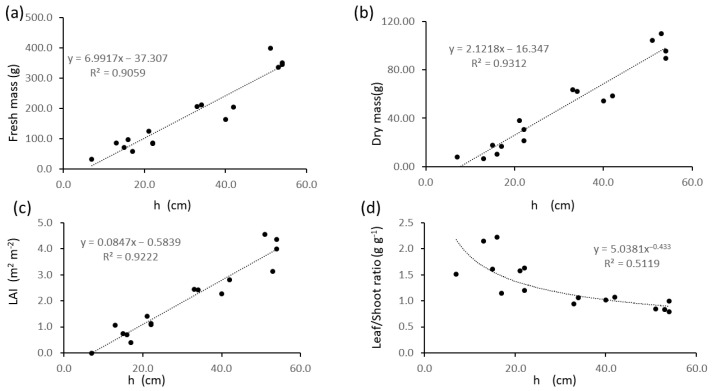
Bivariate plots of alfalfa biometrics as a function of plant height: (**a**) fresh biomass; (**b**) dry biomass; (**c**) Leaf Area Index; (**d**) leaf/stem mass ratio.

**Figure 6 plants-13-01085-f006:**
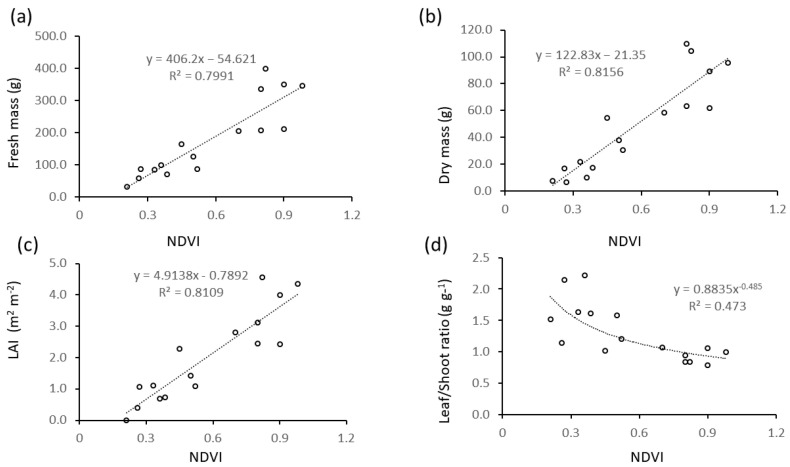
Bivariate plots of alfalfa biometrics as a function of NDVI: (**a**) fresh biomass; (**b**) dry biomass; (**c**) Leaf Area Index; (**d**) leaf/stem mass ratio.

**Figure 7 plants-13-01085-f007:**
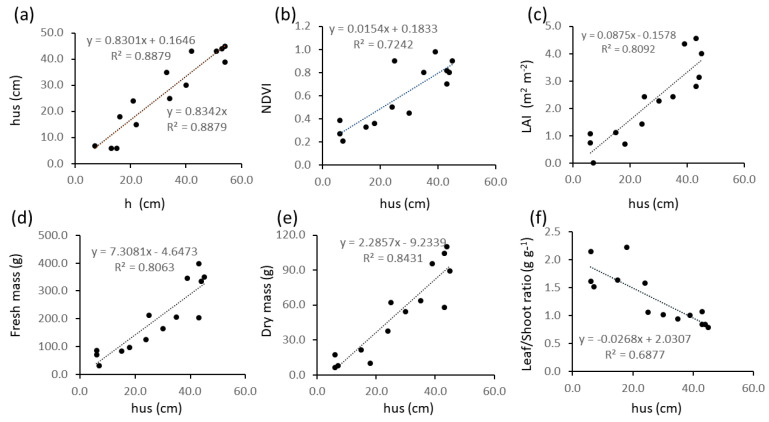
Relationships between alfalfa biometrics and ultrasound-measured plant height (hus): (**a**) hus as a function of manually measured plant height (h); (**b**) NDVI as a function of hus; (**c**) Leaf Area Index as a function of hus; (**d**) fresh mass as a function of hus; (**e**) dry mass as a function of hus; (**f**) leaf to stem mass ratio as a function of hus.

**Figure 8 plants-13-01085-f008:**
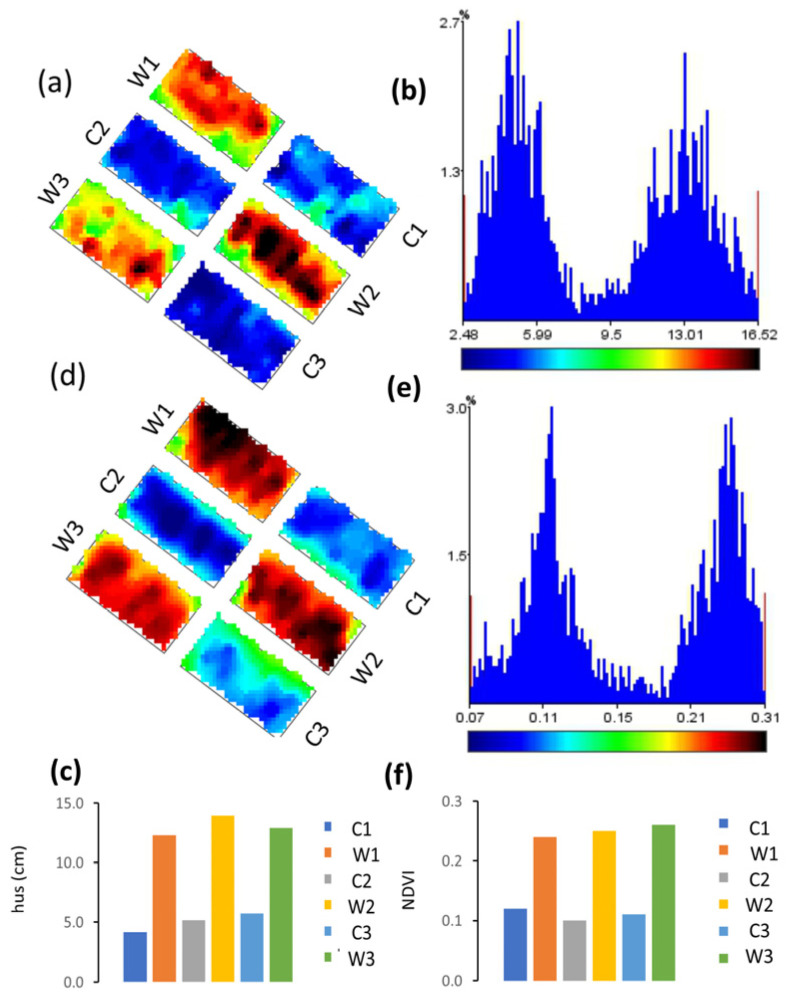
Maps of the experimental field of wheat; (**a**) hus values across the field; (**b**) hus values frequency distribution; (**c**) hus in each plot: colored bars = average values. Line bars: standard deviation; (**d**) NDVI values across the field; (**e**) NDVI values frequency distribution; (**f**) NDVI in each plot: colored bars = average values. Line bars: standard deviation. C1, C2, and C3 = three replicate plots of treatment C = wheat cut at stage 30 of the Zadoks scale; W1, W2, W3 = three replicate plots of treatment W: uncut wheat plants.

**Figure 9 plants-13-01085-f009:**
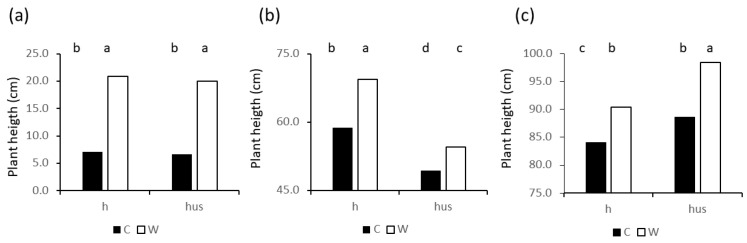
Height of wheat plants from manual measurements (h) or derived from an ultrasound sensor (hs). Values represent averages of treatments C = wheat cut at stage 30 of the Zadoks scale and W: uncut wheat plants. (**a**): the end of tillering; (**b**) booting; (**c**) grain filling. Different letters indicate significant (*p* < 0.05) differences in Tukey’s *post-hoc* test.

**Figure 10 plants-13-01085-f010:**
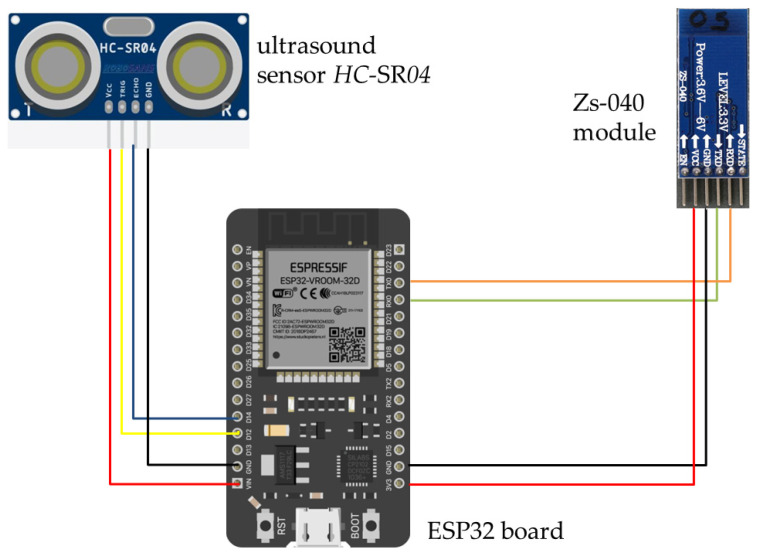
Diagram of connections between components of the platform.

**Figure 11 plants-13-01085-f011:**
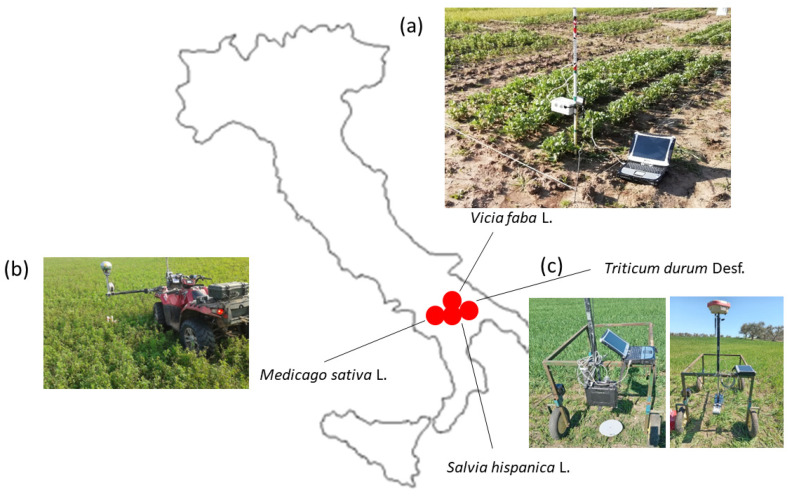
Sites of field measurements and sensor modes of data acquisition: (**a**) pole-mounted for static measurements in a faba bean filed; (**b**) mounted on a quad for on-the go or static measuremts in an alfalfa field; (**c**) mounted on a wheeled chassis for on-the-go measurements in wheat plots. The latter setting was used in a chia field.

**Table 1 plants-13-01085-t001:** Plant biometrics and sensor indices in the ground-truth areas for *Medicago sativa* L. St dev = standard deviation; CV% = coefficient of variation; h = plant height; hus = ultrasound-measured plant height.

	LAI of Alfalfa	Fresh Biomass	Dry Biomass	NDVI	Leaf/Total Mass Ratio	h	hus
	(m^2^ m^−2^)	(g m^−2^)	(g m^−2^)		(g g^−1^)	(cm)	(cm)
Min	0	128.00	26.40	0.21	0.44	13.02	6.11
Max	4.56	1592.00	440.00	0.98	0.69	54.28	45.07
Mean	2.03	714.25	196.65	0.57	0.55	32.47	27.14
St dev	1.44	479.58	143.55	0.26	0.08	15.56	14.84
CV%	47.91	67.14	73.00	45.92	14.44	47.91	54.66

Values showed a high variability: the coefficient of variation ranged between 45.82% of NDVI and 73.00% of plant dry mass, with the exception of the leaf/total mass ratio where values were lower than 15%.

**Table 2 plants-13-01085-t002:** Plant biometrics and sensor indices in *Triticum durum* Desf. transects. hus = ultrasound-measured plant height; NDVI = normalized difference vegetation index. LAI = Leaf area Index. St dev = standard deviation; CV% = coefficient of variation. Summary statistics are calculated on all data across the experimental field.

	hus	h	NDVI	LAI
	(cm)	(cm)		(m^2^ m^−2^)
Min	5.53	6.02	0.10	0.32
Max	24.57	24.92	0.26	2.02
Mean	13.68	14.37	0.18	1.02
St dev	7.02	7.24	0.08	0.61
CV%	51.27	50.41	42.89	60.19

The variability of measurements across the whole experimental field, regardless of treatments, was quite high, with coefficients of variation ranging from 42.89% for NDVI to 60.19% for LAI. Significant (*p* < 0.05) linear regressions are found between h and hus (hus = 0.9638 h − 0.1637 R^2^ = 0.99), and between hus and the leaf area index (LAI = 0.0731 hus + 0.034 R^2^ = 0.79) and the normalized difference vegetation index (NDVI = 0.0103 hus + 0.042 R^2^ = 0.98).

## Data Availability

Data are contained within the article.

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
