# Peer review of "Monitoring Plant Height and Spatial Distribution of Biometrics with a Low-Cost Proximal Platform"

_plants, 2024, doi:10.3390/plants13081085_

Round 1

Reviewer 1 Report

Comments and Suggestions for Authors

This paper entitled “Monitoring plant height and spatial distribution of biometrics with a low-cost proximal platform” is well-written and the discussion is clear and complete. Please find some remarks below:

1. Please add the explanation of landraces vm1-vm4 in Figure 2. (b) and (c).

2. Please increase the resolution of Figure 3. It is hard to read this figure.

3. LN 137-139 Why the bare-soil points cause this result?

4. LN 318-319 What do you mean about these sentences?

Author Response

We thank the reviewer for suggestions and have revised by addressing comments as specified in the following point-by-point responses. 

Resposes to Reviewer 1:

This paper entitled “Monitoring plant height and spatial distribution of biometrics with a low-cost proximal platform” is well-written and the discussion is clear and complete. Please find some remarks below:

  1. Please add the explanation of landraces vm1-vm4 in Figure 2. (b) and (c).

The explanation was added in Materials and methods : “ For the botanical variety Vicia faba (L.) var. maior (Harz) Beck, we used the commercial variety Aguadulce and accessions sourced from the germplasm collection of the Institute of Biosciences and Bioresources (IBBR) of the Italian National Research Council (CNR) in Bari, Italy:

  • Vma1= Accession number 112906 from USA Vma1
  • Vma2 = Accession number 103235 from Italy
  • Vma3 = Accession number 107620 from Greece.
  • Vma4 = Accession number 106374 from Algeria

For the botanical variety Vicia faba L. var. minor (Harz) Beck we used the commercial variety Prothabat and accessions sourced from the germplasm collection of the Institute of Biosciences and Bioresources (IBBR) of the Italian National Research Council (CNR) in Bari, Italy:

  • Vmi1 = Accession number 113620 from Germany
  • Vmi 2= Accession number 113620 from Germany
  • Vmi3 = Accession number 109322 from Ethiopia
  • Vmi4 = Accession number 118952 from Afghanistan

  1. Please increase the resolution of Figure 3. It is hard to read this figure.

We increased readability of figure 3 by making lines finer and more definite and by saving at a much higher resolution.

  1. LN 137-139 Why the bare-soil points cause this result?

We added the following text to the “Discussion” section about this point:

“In Salvia hispanica (Fig. 3) hus values closely reproduced plant height except for at the edges of measured transect. In this experiment the edges of transects corresponded to plot edges, therefore to regions where bare soil and/or segetal species were found, with height different from chia. We hypothesize that in this case specific edge effects coupled with shifts and systematic errors in sensor measurements we recorded in this dataset may be due to an imperfect perpendicularity of the sensor, which may have picked up reflections from bare soil at the beginning of the transect, thus underestimating plant height, and reflections from the crop at the transect end, thus overestimating plant height.”   

  1. LN 318-319 What do you mean about these sentences?

We reworded as: “After quantifying misalignments, we used cross-correlation or dynamic time-warping distance (DTW) to shift hus measurements one of one or few lags and obtained a better correspondence of sensor and measured data (e.g. Fig. 3 c-d for chia plants spaced at 30 cm). “

Reviewer 2 Report

Comments and Suggestions for Authors

The height of crops is influenced by many factors, especially varieties, local climate, etc. However, we cannot determine whether the height of crops has a good or bad impact on the ecosystem and crop yield. Many crops we like are tall, while others we like are short. So I feel that the research significance and hypothesis of this article are not valid. In addition, there are many methods and even the platform used in this article that can measure the height of crops, so there is no need to create another platform specifically for crop height.

Perhaps forests or trees require such a platform.

Author Response

We thank the reviewer for suggestions and have revised by addressing comments as specified in the following point-by-point responses. 

Resposes to Reviewer 2:

The height of crops is influenced by many factors, especially varieties, local climate, etc. However, we cannot determine whether the height of crops has a good or bad impact on the ecosystem and crop yield. Many crops we like are tall, while others we like are short. So I feel that the research significance and hypothesis of this article are not valid. In addition, there are many methods and even the platform used in this article that can measure the height of crops, so there is no need to create another platform specifically for crop height.

Perhaps forests or trees require such a platform.

We agree that plant height is affected by many factors and that is why a large body of literature identifies height as an important indicator of plant status and a good predictor of future yield. For some processes it also goes beyond that and is a proxy of biomass and therefore carbon stock,  and has a specific role e.g. for plant competition. This is why specific crop height measurements are targeted by plant research. We improved our introduction by adding:

“ Plant height is the result of many genetic, environmental and management factors…”

and we argued our platform is useful by adding:

“Compared to LiDAR, ultrasonic sensors are less expensive and do not involve possible harm to the naked eye. LiDAR, on the other hand, can make measurements at a larger distance and features narrow point detection; this can be a pro since it gives a higher accuracy but also a con since given objects may be missed in a prospection “

and:

“We aimed at devising a low-cost system for measuring plant height which could be used in a flexible way, both on-the-go and in static mode, and at proving its usefulness in different  applications from single plants to plots and whole fields. We therefore aimed at testing the device across a range of vegetation types, from crop row transects to whole-field mapping. “

The last paragraphs of the “Discussion” section clarify that our platform, though specific for crop height, can be integrated with other sensors in order to measure crops height as long as other traits,

“The board used for our sensor is especially amenable for flexible uses and therefore to address error sources linked to different settings [e.g. 39] and for to use for in mono-sensors or for multi-sensor platform (e.g. [40]) Internet of Things applications given the different connectivity modes from Wi-Fi to Bluetooth Classic and Bluetooth Low Energy (BLE), Sim card or Ethernet support and additional interfaces, including UART, SPI, I2C, and ADC, enabling connection to a wide range of peripheral devices. Flexibility is also given by a high number of General-Purpose Input/Output pins and a large flash memory for programming and storage, and compatibility with other boards and development platforms..”

We added a “Conclusions” section where we  outline the main advantage of our platform as the ease of use and flexibility for measurements in different settings and by users with different skills and inclination for technology. This section was written in a “highlight” mode with short sentences highlighting the main issues and future research needs:

“5. Conclusions

We devised and tested a low-cost platform for measuring plant height with an ultrasound sensor designed to be used in different modes, from static to on-the go.

We generated point measurements with static setting on faba bean, with a high regression coefficient where the ultrasound sensor proved able to repro-duce height regardless of genotype and phenological stage, thus showing the potential of the platform for static single-plant measurements.

High regression coefficients were also shown in the on-the-go mode for settings with a quite continuous plant cover such as narrow-spaced chia, wheat and alfalfa, whereas sensor data were not able to closely reproduce crop height where sharp variations were found such as in widely-spaced chia.

We were able to quantify and correct some systematic errors such as data misalignment with feature-based (cross correlation) or shape-based (dynamic time warping) measures of similarity between data series.

Ultrasound-measured plant height proved to be a better predictor than NDVI for plant biometrics relevant for water relations and yield behavior such as Leaf Area Index and biomass in alfalfa.

Overall ultrasound-measured height with our platform proved to be a fast and low-cost method of estimating crop parameters useful in ecological research or agriculture applications.

Important characteristic of our platform are that it is simple and flexible given the possibility to be employed by users with different skills and inclination for technology and in different settings, from mounted on simple poles for single-point static measurements, to towed manually for the complete characterization of plots or rows in experiments. Further the platform may be carried on vehicles for mapping large surfaces like open fields, prairies and natural herbaceous vegetation sites providing maps useful for spatial management and characterization of spatial varia-tion of plant responses in different conditions from natural to managed systems.

One of the features of the platform’s ESP32 board is wide connectivity, therefore future developments may include the design of a multi-sensor platform with the same flexibility of use in different settings.

Also future work should focus on the analysis and correction of errors linked to field settings and to modes of platform use such as misalignment of data.”

Reviewer 3 Report

Comments and Suggestions for Authors

1.introduction should address the problem that this paper solved. Why only ultrasonic sensor is the low cost? Any other methods also are low cost?

2. The biometrics parameters should be also addressed in the experimental scheme. The scheme are nor clear now.

3. Dynamic Time Warping normalized Distance (DTW) is not appear in the experiments and results. Only in conclusions.

4. How many cost did the low cost platform?

5. What is the final conclusions with low cost platform in different plants?

6. The conclusions should be improved with more highlights, and the future works should also addressed and extended. 

Author Response

We thank the reviewer for suggestions and have revised by addressing comments as specified in the following point-by-point responses. 

Resposes to Reviewer 3:

1.introduction should address the problem that this paper solved. Why only ultrasonic sensor is the low cost? Any other methods also are low cost?

We did not stress the issue of comparing prices between sensors, because prices vary considerably between sources (e.g. different vendors or websites) and change in time therefore we can state our sensor is low cost but this doesn’t mean other sensors are expensive.  Among sensors specific for height, though, the price of  LIDAR sensors has gone down considerably but still ultrasonic sensors are cheaper.

We therefore improved the introduction by adding:

“Compared to LiDAR, ultrasonic sensors are less expensive and do not involve possible harm to the naked eye. LiDAR, on the other hand, can make measurements at a larger distance and features narrow point detection; this can be a pro since it gives a higher accuracy but also a con since given objects may be missed in a prospection.”

and we stated our objectives better:

“We aimed at devising a low-cost system for measuring plant height which could be used in a flexible way, both on-the-go and in static mode, and at proving its usefulness in different  applications from single plants to plots and whole fields. We therefore aimed at testing the device across a range of vegetation types, from crop row transects to whole-field mapping. “

  1. The biometrics parameters should be also addressed in the experimental scheme. The scheme are nor clear now.

Regarding biometrics and radiometrics we changed the “Materials and methods” section by:

  • Adding the following text at the beginning of the tests description:

“In all tests the height of target objects or vegetation from ultrasound measurements (hus) was calculated as :

hus = hs – usd (cm)

where

  hs = height of sensor from the ground

usd = ultrasound-measured distance between target objects or vegetation and sensor.”

  • Adding a description of the other biometric and radiometric measurements which we structured in numbered paragraphs for clarity in the alfalfa section and referred to it for wheat:

            “The following plant biometric and radiometric measurements were made:

4.3.4.1 NDVI

A radiometric Greenseeker ® (Trimble Sunnyvale, CA, U.S.A.) sensor was used to meaasure reflected radiation in the red ( ~660 nm) and near infrared (~770 nm) wavelengths for the calculation of the Normalized Differences Vegatation Index (NDVI):

NDVI = (NIR – VIS) /(NIR)+(VIS)

Where

NIR = reflectance in the infrared band (~770 nm)

VIS = reflectance in the red band (~660 nm).

4.3.4.2 Leaf Area Index

Leaf area index (LAI m2 m-2) was measured with a LI-COR 2200c (LI-COR Lincoln, Nebraska USA) field leaf-area meter.

4.3.4.3 Vegetation height

Vegetation height (h) was measured with a rigid measuring tape as the distance from the ground of a 10-cm diameter disk mounted to a stick and placed on top of the canopy every 0.1 m on a 0.5 x 0.5 m area.

4.3.4.4 Biomass

 Above-ground plant parts were harvested on 0.5 x 0.5 m areas and weighed fresh and after oven-drying at 65°C until constant weight. “

  1. Dynamic Time Warping normalized Distance (DTW) is not appear in the experiments and results. Only in conclusions.

Dynamic Time Warping normalized Distance or DTW also appears

in “Materials and Methods” at lines 582-3

in “Results” at lines 100, 137, 143, 150,  156, 158

in “Discussion” at lines 283-4, 297-8, 306.

Dynamic Time Warping also appears at lines 21, 26, 100, 122, 125, 128, 282, 293, 303, 600

  1. How many cost did the low cost platform?

In “Materials and methods” we added cost to all platform components. There is a wide variation of prices depending on the source, therefore we indicated a range (e.g. from specialized asian websites to Amazon):

The current cost of an ESP32 board ranges from 1.75 to 8 Euros depending on the source.

The current cost of an HC-SR04 ranges from 2 to 10 Euros depending on the source.

The current cost of a Zs-040 module ranges from 0.3 to 10m Euros depending on the source.

Protective plastic case (current cost 5 Euros).

Therefore the sensor platform costs from 9.05 to 32 Euros

We also added the cost of an extra component which may be used if the device is not powered through the computer USB port: A 7860mAh power bank: current cost 15 Euros.

  1. What is the final conclusions with low cost platform in different plants?
  2. The conclusions should be improved with more highlights, and the future works should also addressed and extended.

We added a “Conclusions” section where we drew conclusions for the different plants and settings and moved the overall conclusions. Theis section was written in a “highlight” mode with short sentences highlighting the main issues and future research needs:

“5. Conclusions

We devised and tested a low-cost platform for measuring plant height with an ultrasound sensor designed to be used in different modes, from static to on-the go.

We generated point measurements with static setting on faba bean, with a high regression coefficient where the ultrasound sensor proved able to repro-duce height regardless of genotype and phenological stage, thus showing the potential of the platform for static single-plant measurements.

High regression coefficients were also shown in the on-the-go mode for settings with a quite continuous plant cover such as narrow-spaced chia, wheat and alfalfa, whereas sensor data were not able to closely reproduce crop height where sharp variations were found such as in widely-spaced chia.

We were able to quantify and correct some systematic errors such as data misalignment with feature-based (cross correlation) or shape-based (dynamic time warping) measures of similarity between data series.

Ultrasound-measured plant height proved to be a better predictor than NDVI for plant biometrics relevant for water relations and yield behavior such as Leaf Area Index and biomass in alfalfa.

Overall ultrasound-measured height with our platform proved to be a fast and low-cost method of estimating crop parameters useful in ecological research or agriculture applications.

Important characteristic of our platform are that it is simple and flexible given the possibility to be employed by users with different skills and inclination for technology and in different settings, from mounted on simple poles for single-point static measurements, to towed manually for the complete characterization of plots or rows in experiments. Further the platform may be carried on vehicles for mapping large surfaces like open fields, prairies and natural herbaceous vegetation sites providing maps useful for spatial management and characterization of spatial varia-tion of plant responses in different conditions from natural to managed systems.

One of the features of the platform’s ESP32 board is wide connectivity, therefore future developments may include the design of a multi-sensor platform with the same flexibility of use in different settings.

Also future work should focus on the analysis and correction of errors linked to field settings and to modes of platform use such as misalignment of data.”

Round 2

Reviewer 2 Report

Comments and Suggestions for Authors

Authors have addressed all reviewers' comments.

Reviewer 3 Report

Comments and Suggestions for Authors

The paper was revised well.